# An Analysis of Characteristics of Post-Stroke Fatigue in Patients without Depression: A Retrospective Chart Review

**DOI:** 10.3390/brainsci11121642

**Published:** 2021-12-13

**Authors:** Yu Jin Lee, Woo-Sang Jung, Seungwon Kwon, Chul Jin, Seung-Yeon Cho, Seong-Uk Park, Sang-Kwan Moon, Jung-Mi Park, Chang-Nam Ko, Ki-Ho Cho

**Affiliations:** Department of Cardiology and Neurology, College of Korean Medicine, Kyung Hee University, Seoul 02447, Korea; yjl1992@naver.com (Y.J.L.); wsjung@khu.ac.kr (W.-S.J.); yahaly@naver.com (C.J.); sy.cho@khu.ac.kr (S.-Y.C.); seonguk.kr@gmail.com (S.-U.P.); skmoon@khu.ac.kr (S.-K.M.); pajama@khu.ac.kr (J.-M.P.); kcn202@khu.ac.kr (C.-N.K.); kihocho58@gmail.com (K.-H.C.)

**Keywords:** post-stroke fatigue, depression, apolipoprotein A1, pattern identification, retrospective chart review study

## Abstract

Post-stroke fatigue (PSF) is among the most common stroke sequelae and affects rehabilitation, resulting in poor recovery. A main influencing factor may be depression, which has been investigated with fatigue in several clinical trials. We aimed to evaluate the characteristics of fatigue in post-stroke patients without depression through a retrospective chart review. The medical records of stroke patients hospitalized in the Stroke and Brain Disease Center, Kyung Hee University Korean Medicine Hospital were reviewed. Stroke patients without depression were divided into a PSF group and control group (without fatigue). The demographic characteristics, type of stroke, medical history, laboratory examinations, clinical features, and pattern identification of each patient were recorded and compared between the study groups. The medical records of 216 patients were reviewed; 85 and 131 patients were assigned to the PSF and control group, respectively. Apolipoprotein A1 levels were significantly lower in the PSF than in the control group (105.6 ± 16.5 vs. 116.2 ± 21.8). We found a significantly higher occurrence of reversal cold of the extremities and a lower probability of fire-heat pattern in the PSF group than in the control group. This study suggests that apolipoprotein A1 levels are lower and cold manifestations are more common in PSF patients without depression than in those without fatigue.

## 1. Introduction

Stroke, a result of disruption to the cerebral blood flow, is one of the most common neurological disorders. It is the third most prevalent cause of death in Korea, with 42.7% of cases occurring in males and 46.1% in females [1]. Aside from its high mortality rate, stroke leads to severe sequelae after survival, including physical, cognitive, mood, and behavioral disorders [2]. Post-stroke fatigue (PSF) is a highly complex motor-perceptual, psychological, and cognitive experience that stroke victims continue to face [3] and it is a common mood disorder following stroke, with a prevalence of 16–85% [4,5,6,7] in stroke patients. Fatigue is reported to negatively affect post-stroke rehabilitation and recovery [8]. Post-stroke fatigue has been linked to a decline in stroke patients’ physical condition, a decrease in self-efficacy during physical activity, a deterioration in rehabilitation treatment outcomes, a decrease in social participation, a decrease in quality of life, functional limitations, and an increase in mortality [7].

While PSF is known to have a high incidence rate and a negative influence on the prognoses of stroke patients, studies on other possible influencing factors are still in progress due to the lack of elucidation of the PSF pathophysiology. A review article on the characteristics of PSF [7] discusses its association with factors such as age and sex, type of stroke, autonomic nervous system abnormalities, inflammation, and neuroendocrine control. However, the inconsistent results among studies show that the relationships remain inconclusive.

Furthermore, clinical conditions, such as movement disorders, sleep disorders, and depression, have been suggested to be associated with fatigue. In particular, depression is discussed in multiple studies as one of the primary factors influencing fatigue [9,10,11,12]. However, due to difficulties in separately assessing fatigue and depression, along with the strong association between the two conditions [13], the influencing factors and characteristics of fatigue alone are not well known. In previous studies on the biomarkers for PSF, the disorder has been associated with blood glucose [14] and inflammatory markers [15,16]. However, in both studies, these markers have also been observed to be associated with depression.

Although 50% of stroke patients without depression complain of fatigue [12], previous studies have not separately evaluated fatigue and depression. Thus, this study aimed to evaluate the characteristics of fatigue alone in post-stroke patients without concomitant depression. By studying the differences in pattern identification based on the reported symptoms, this study further aimed to contribute to effectively treating PSF in the future. This paper reports the novel findings obtained through conducting the study as outlined below.

## 2. Methods

### 2.1. Study Subjects

#### 2.1.1. Inclusion Criteria

The following inclusion criteria were used to review the medical records of patients who did not have depression at the time of their inpatient stroke therapy.

(1) Aged 19 years or above and hospitalized in the Stroke and Brain Disease Center, Kyung Hee University Korean Medicine Hospital, between 1 March 2019 and 3 December 2020 for treatment of disorders classified under code I63 (cerebral infarction) or I61 (intracerebral hemorrhage) of the Korean Standard Classification of Diseases and Causes of Death.

(2) Assessed to be non-depressed, with a total score of 9 or less on the Patient Health Questionnaire-9 (PHQ-9) at the time of hospitalization.

(3) Answered the Fatigue Assessment Scale (FAS) and Fatigue Severity Scale (FSS) questionnaires, have been checked for fatigue awareness, and have undergone blood tests at the time of hospitalization.

#### 2.1.2. Exclusion Criteria

The medical records of patients who had the following characteristics were excluded from the review:

(1) Had unverifiable medical records for the satisfaction of the aforementioned inclusion criteria.

(2) Initially diagnosed with stroke, but final diagnosis changed to one other than cerebral infarction or cerebral hemorrhage upon complete examination.

### 2.2. Study Design

A retrospective study was conducted through the investigation of medical records, following Institutional Review Board (IRB) approval from Kyung Hee University Korean Medicine Hospital (KOMCIRB 2021-05-002). The present study was a study in which information that does not contain personally identifiable information of subjects was received from the hospital and analyzed. Therefore, informed consent was not required for each subject. This consent waiver was approved by the IRB. In addition, the present study was conducted in compliance with clinical research guidelines, such as the Declaration of Helsinki. The protocol of this study was registered with the Clinical Research Information Service (CRIS) under number KCT0006328.

After the selection of patients using the inclusion and exclusion criteria, the study subjects were classified into the post-stroke fatigue (PSF) group if they had two or more of the following: (1) presence of subjective symptoms of fatigue, (2) a total score ≥ 24 on the FAS, and (3) a total score of ≥4 on the FSS. Otherwise, they were classified into the non-post-stroke fatigue (non-PSF) group.

The medical records of each patient assigned to a group were assessed for the collection of the following data: (1) demographic characteristics, (2) stroke-related characteristics, (3) laboratory examination results, and (4) clinical features and pattern identifications.

### 2.3. Parameters

The following information was collected through access to the medical records of the study subjects.

#### 2.3.1. Demographic Characteristics

(1)Age.(2)Sex.(3)Body mass index (BMI).(4)Religious status.(5)Smoking status.(6)Drinking status.

#### 2.3.2. Stroke-Related Characteristics

(1)Duration of stroke.(2)Type of stroke: ischemic stroke, hemorrhagic stroke.(3)Assessment of neurologic damage: National Institutes of Health Stroke Scale (NIHSS) score.(4)Assessment of cognitive function: Korean version of the Mini-Mental State Examination (MMSE-K) score.(5)Assessment of movement function: Modified Barthel Index (MBI), Manual Function Test (MFT) scores.(6)Medical history of stroke and various risk factors: medical histories of stroke, hypertension, dyslipidemia, diabetes mellitus, heart disease, and cancer were collected as listed on patients’ initial hospitalization records.

#### 2.3.3. Fatigue Assessment

(1) FAS

On admission, the patient was requested to fill out the FAS. An FAS is a self-report questionnaire that contains 10 questions on physical and psychological fatigue [17].

(2) FSS

On admission, the patient was requested to fill out the FSS, a fatigue-related, nine-item self-report questionnaire with a single dimension [18].

(3) Subjective perception of fatigue

Patients were asked about fatigue awareness at the time of admission, and the findings of the survey were used.

#### 2.3.4. Laboratory Testing Results

(1) Complete Blood Count

Data on the patient’s white blood cells; segments of lymphocytes, monocytes, eosinophils, basophils, neutrophils; red blood cells; hemoglobin; platelet; erythrocyte sedimentation rate (ESR); prothrombin time; international normalized ratio; and activated partial thromboplastin time (aPTT) were collected from a complete blood count (CBC) performed at the time of hospitalization.

(2) Biochemical Tests

Data on the patient’s total protein, albumin, total bilirubin, glucose, blood urea nitrogen, creatinine, aspartate aminotransferase, alanine aminotransferase, alkaline phosphatase, phosphorus, calcium, sodium, potassium, chloride, uric acid, gamma-glutamyl transferase, creatine kinase, C-reactive protein (CRP), and high sensitivity CRP (hs-CRP) were retrieved from the initial biochemical tests performed at the time of hospitalization.

(3) Blood Lipid Tests

Data on the patient’s total cholesterol, triglyceride, low-density lipoprotein (LDL), high-density lipoprotein, apolipoprotein A1, apolipoprotein B, total lipid, and phospholipid were collected from lipid analyses performed at the time of hospitalization.

(4) Endocrinologic Tests

Data on homocysteine, hemoglobin A1c, and thyroid stimulating hormones were collected from endocrinological tests performed at the time of hospitalization.

#### 2.3.5. Clinical Features and Pattern Identification

Clinical features that could be confirmed from the pattern identification records initially taken upon hospitalization, including insomnia, complexion, faint low voice, thirst, reversal cold of the extremities, and pulse type, were noted. The likelihood of each pattern based on the data was calculated using the pattern identification regression equation developed by Jung et al. [19] for stroke patients.

### 2.4. Statistical Analysis Methods

For the identification of the differences in characteristics between the study groups, *t*-tests were performed for continuous variables that followed the normal distribution, Mann–Whitney U tests were performed for continuous variables that did not follow the normal distribution, and chi-square tests were performed for categorical variables. To determine the factors that affect PSF, a multiple logistic regression analysis was performed on the statistically significant variables in between-group comparisons with *p*-values ≤ 0.05. IBM SPSS ver. 25 (IBM Co., Armonk, NY, USA) was used for the statistical analysis.

## 3. Results

A search of the medical records found 502 patients who were hospitalized for treatment in the Stroke and Brain Disease Center, Kyung Hee University Korean Medicine Hospital, between 1 March 2019, and 3 December 2020, under the primary diagnosis of code I639 or I619. Among them, 175 patients who had insufficient data on PHQ-9, FAS, FSS, or fatigue awareness questionnaires, and 111 patients who were classified as depressed, with scores of ≥10 on the PHQ-9, were excluded. The medical records of the 216 remaining patients were included in the analysis. Based on the group classification criteria, 85 were assigned to the PSF group and 131 were assigned to the non-PSF group (Figure 1).

### 3.1. Comparison of Demographic Characteristics

The sex ratio, BMI, religious status, proportion of smokers, and proportion of drinkers did not show significant differences between the two groups. The average age was significantly different between the two groups (*p* = 0.024), with 68.0 ± 14.2 years and 64.6 ± 11.9 years in the PSF and non-PSF groups, respectively (Table 1).

### 3.2. Comparison of Stroke-Related Characteristics

The duration of stroke did not show a significant difference between the two groups. The proportion of hemorrhagic stroke was higher in the PSF group (16.5%) than in the non-PSF group (7.6%). The levels of neurologic damage, cognitive function, and movement function, as assessed by the NIHSS, MMSE-K, and MBI and MFT, respectively, did not show significant differences between the two groups. Similarly, the medical histories of stroke, hypertension, dyslipidemia, diabetes mellitus, heart disease, and cancer did not show significant differences between the two groups (Table 2).

### 3.3. Comparison of Laboratory Examination Results

The CBC results showed that the PSF group had a significantly higher ESR and a longer aPTT than the non-PSF group (Table 3). No variable in the biochemical tests was found to have a significant difference between the groups (Table 4). The blood lipid profiles showed that the level of apolipoprotein A1 was significantly lower in the PSF group than in the non-PSF group, with values of 105.6 ± 16.5 mg/dL and 116.2 ± 21.8 mg/dL, respectively (Table 5, Figure 2). No variable in the endocrinological tests was found to have a significant difference (Table 5).

### 3.4. Comparison of Clinical Features and Pattern Identification

From the collection of symptoms reported by each study subject, a significant difference between the two groups was found in the presence of reversal cold of the extremities, with 17 (20.0%) and 13 (9.9%) patients in the PSF and non-PSF groups, respectively. No significant differences in other clinical features, including insomnia, complexion, faint low voice, thirst, and pulse type, were observed between the groups. Calculations using the pattern identification regression equation showed that there was a significantly lower likelihood of fire-heat pattern in the PSF group than in the non-PSF group (Table 6).

### 3.5. Multivariate Analysis on Factors Affecting PSF

To identify the independent effect of each factor on PSF, a multiple logistic regression analysis was performed using backward elimination on factors that were shown to be significant in the between-group comparisons. Apolipoprotein A1 and fire-heat pattern were found to have significant negative effects, at a significance level of *p* < 0.05 (Table 7).

## 4. Discussion

To understand the characteristics of PSF patients without depression, this study analyzed the demographic characteristics, stroke-related characteristics, blood test results, and symptomatic complaints of patients who had been hospitalized in the Stroke and Brain Disease Center, Kyung Hee University Korean Medicine Hospital. The study included 216 stroke patients who were not diagnosed with depression, among whom 85 (39.4%) were assigned to the PSF group. The ratio between the groups was within the acceptable range, and was shown to be similar to the 50% ratio in a previous study conducted in Asan Medical Center investigating the characteristics of stroke patients [11].

The between-group comparisons showed that the apolipoprotein A1 level was lower in the PSF group than in the non-PSF group. In addition, the multivariate regression analysis indicated a negative effect of apolipoprotein A1 on fatigue. Apolipoprotein A1 is the primary protein constituting high-density lipoprotein, and can pass through the blood–brain barrier due to its small molecule size. In acute ischemic stroke, serum apolipoprotein A1 has been used as a predictive indicator [20]. Furthermore, reports have suggested that it controls Aβ aggregation and toxicity, such as Aβ-induced nerve damage, through the combination with Aβ (amyloid-beta peptide) [21]. Temporary increases in Aβ plaques and cerebral amyloid angiopathy are known to be present around the site of the lesion in strokes [22]. As there is a study suggesting an association between Aβ and fatigue [23], a possible mechanism could be considered where blood apolipoprotein A1 levels affect the Aβ toxicity inside the brain, leading to differential complaints of fatigue in post-stroke patients. Further prospective studies need to be conducted for confirmation.

There were significantly more complaints of reversal cold of the extremities in the PSF group, and while not statistically significant, there were fewer complaints regarding complexion or thirst in the PSF group. This may indicate that there was a higher occurrence of cold syndrome [24] in the PSF group than in the non-PSF group. Furthermore, the likelihood of the fire-heat pattern was observed to be significantly lower in the PSF group than in the non-PSF group. Cold syndrome is known as a collection of symptoms related to declines in energy metabolism and metabolic functions [25]. While this may be related to PSF, insufficient studies have been conducted on this relationship. Kim et al. [26] reported that bojungikgi-tang, ssanghwa-tang, and sipjeondaebo-tang were frequently prescribed for patients with chronic fatigue syndrome or idiopathic chronic fatigue. *Astragalus membranaceus* and *Panax ginseng* in bojungikgi-tang, *Astragalus membranaceus* and *Cinnamomum cassia* in sipjeondaebo-tang, and *Astragalus membranaceus* and *Cinnamomi ramulus* in ssanghwa-tang are warm medicinal herbs widely used for treating cold syndrome [27]. Moreover, among the treatment methods of complementary and alternative medicine, heat therapy is recommended by the Centers for Disease Control and Prevention for selective use to alleviate the symptoms of chronic fatigue syndrome [28]. This shows that the treatments for cold syndrome are currently in clinical use for treating fatigue, which should similarly be applied in PSF patients with accompanying cold syndrome. This implies the importance of the cold–heat pattern in determining the appropriate treatment for complaints of PSF.

This is a single-center study; thus, further efforts should be made to replicate the obtained results in larger populations to generalize the recommendations for all cases of fatigue in stroke patients. Furthermore, because the present study had a limited sample size, the cut-off value for using apolipoprotein A1 or the fire-heat pattern as PSF biomarkers could not be calculated. Additionally, there are limitations in identifying causal relationships due to the observational and cross-sectional design of the study, which require prospective studies for the future identification of causal relationships. Despite having such limitations, this study carries clinical significance as the first study to focus on the characteristics of patients who complained exclusively of PSF, without concomitant depression. The results suggested the possibility that PSF, wherein the causes and pathogenesis are fully elucidated, may be associated with blood lipoproteins. Based on the review of patient complaints, the association between PSF and the cold–heat pattern has also been observed, which may be used as a basis for determining the instructions of future PSF treatments in Korean medicine.

## Figures and Tables

**Figure 1 brainsci-11-01642-f001:**
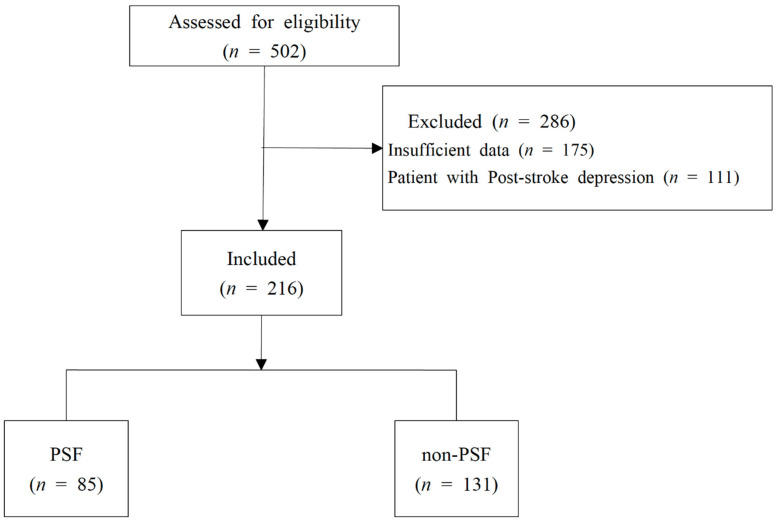
Flowchart of patient inclusion. PSF group: patients with post-stroke fatigue. Non-PSF group: patients without post-stroke fatigue.

**Figure 2 brainsci-11-01642-f002:**
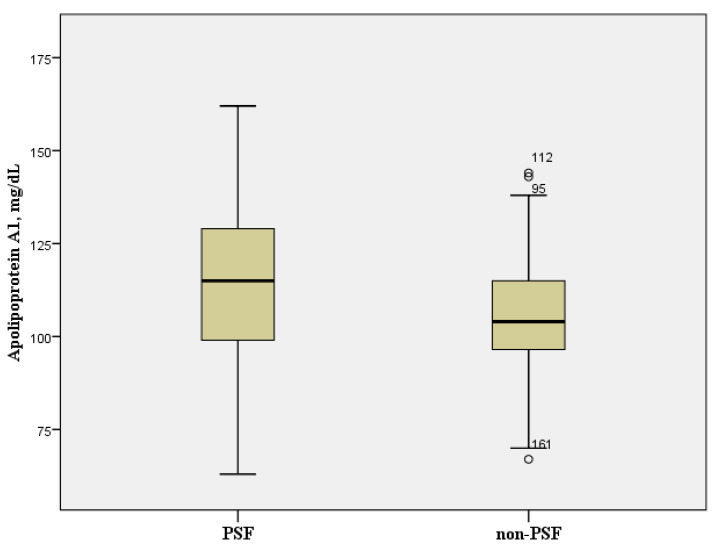
Box plot of apolipoprotein A1. PSF group: patients with post-stroke fatigue. Non-PSF group: patients without post-stroke fatigue.

**Table 1 brainsci-11-01642-t001:** Comparison of demographic characteristics between the PSF and non-PSF groups.

	PSF (*n* = 85)	Non-PSF (*n* = 131)	*p*-Value ^a^
**Age**, *years*	68.0 ± 14.2	64.6 ± 11.9	0.024
**Sex**, *n (%)*			
Male	46 (54.1)	84 (64.1)	0.142
Female	39 (45.9)	47 (35.9)	
**BMI**, *kg/m^2^*	23.58 ± 3.75	24.20 ± 3.48	0.093
**Having religion**, *n (%)*			
Yes	45 (52.9)	55 (42.3)	0.126
No	40 (47.1)	75 (57.7)	
***Smoking***, *n (%)*			
Smoker	27 (31.8)	47 (35.9)	0.534
Non-smoker	58 (68.2)	84 (64.1)	
***Alcohol consumption***, *n (%)*			
Drinker	22 (25.9)	44 (33.6)	0.230
Non-drinker	63 (74.1)	87 (66.4)	

Values are expressed as mean ± SD or number (%). PSF group: patients with post-stroke fatigue; non-PSF group: patients without post-stroke fatigue; BMI: body mass index. ^a^ Mann–Whitney U test used for age and BMI, and chi-square test used for categorical variables.

**Table 2 brainsci-11-01642-t002:** Comparison of disease characteristics between the PSF and non-PSF groups.

	PSF (*n* = 85)	Non-PSF (*n* = 131)	*p*-Value ^a^
**Disease duration**, *d*	64.4 ± 118.0	190.7 ± 920.2	0.356
**Stroke subtype**, *n(%)*			
Ischemic	71 (83.5)	121 (92.4)	0.043
Hemorrhagic	14 (16.5)	10 (7.6)	
**NIHSS**	4.2 ± 3.9	4.2 ± 4.3	0.918
**MMSE-K**	23.8 ± 5.7	25.3 ± 4.0	0.109
**MBI**	56.6 ± 27.9	60.3 ± 25.5	0.477
**MFT**	18.6 ± 9.6	17.9 ± 10.4	0.889
***Medical history***, *n(%)*			
Stroke	20 (23.5)	22 (16.8)	0.222
HTN	49 (57.6)	60 (45.8)	0.089
Dyslipidemia	22 (25.9)	42 (32.1)	0.331
DM	28 (32.9)	51 (38.9)	0.372
Heart Disease	14 (16.5)	23 (17.6)	0.836
Cancer	7 (8.2)	9 (6.9)	0.708

Values are expressed as mean ± SD or number (%). PSF group: patients with post-stroke fatigue; non-PSF group: patients without post-stroke fatigue; NIHSS: National Institutes of Health Stroke Scale; MMSE-K: Korean version of the Mini-Mental State Examination; MBI: Modified Barthel Index; MFT: Manual Function Test; HTN: hypertension; DM: diabetes mellitus. ^a^
*t*-test used for MBI, Mann–Whitney U test used for other continuous variables, and chi-square test used for categorical variables.

**Table 3 brainsci-11-01642-t003:** Comparison of hematologic parameters between the PSF non-PSF groups.

	PSF (*n* = 85)	Non-PSF (*n* = 131)	*p*-Value ^a^
**WBC**, *10^3^/μL*	6.89 ± 2.39	6.85 ± 2.15	0.666
**RBC**, *10^6^/μL*	4.32 ± 0.49	4.36 ± 0.54	0.563
**Hemoglobin**, *g/dL*	13.42 ± 1.61	13.96 ± 3.20	0.152
**Platelet**, *10^3^/μL*	241.1 ± 76.6	240.1 ± 81.5	0.927
**ESR**, *mm/h*	29.1 ± 18.6	24.3 ± 23.2	0.004
**Segment of Lymphocyte**, *%*	28.35 ± 11.18	27.08 ± 7.52	0.359
**Monocyte**, *%*	5.57 ± 1.18	6.07 ± 1.55	0.123
**Eosinophil**, *%*	3.06 ± 2.37	2.62 ± 2.31	0.198
**Basophil**, *%*	0.48 ± 0.31	0.54 ± 0.70	0.173
**Neutrophil**, *%*	61.69 ± 10.62	61.75 ± 8.81	0.963
**PT INR**	1.07 ± 0.34	1.02 ± 0.16	0.411
**aPTT**, *s*	37.20 ± 5.55	35.41 ± 4.43	0.020

Values are expressed as mean ± SD. PSF group: patients with post-stroke fatigue; non-PSF group: patients without post-stroke fatigue; WBC: white blood cell; RBC: red blood cell; ESR: erythrocyte sedimentation rate; PT INR: prothrombin time international normalized ratio; aPTT: activated partial thromboplastin time. ^a^
*t*-test used for RBC, hemoglobin, platelet, lymphocyte, and neutrophil, and Mann–Whitney U test used for other continuous variables.

**Table 4 brainsci-11-01642-t004:** Comparison of biochemical data between the PSF and non-PSF groups.

	PSF (*n* = 85)	Non-PSF (*n* = 131)	*p*-Value ^a^
**Total protein**, *g/dL*	7.00 ± 0.61	6.99 ± 1.02	0.904
**Albumin**, *g/dL*	4.07 ± 0.4	4.47 ± 3.53	0.319
**Total bilirubin**, *mg/dL*	0.69 ± 0.27	1.05 ± 3.68	0.831
**Glucose**, *mg/dL*	127.6 ± 54.1	116.5 ± 41.2	0.608
**BUN**, *mg/dL*	17.3 ± 7.1	16.9 ± 7.2	0.497
**Creatinine**, *mg/dL*	0.84 ± 0.35	0.89 ± 0.66	0.705
**AST**, *U/L*	27.8 ± 15.7	28.8 ± 14.4	0.652
**ALT**, *U/L*	27.7 ± 35.5	27.8 ± 18.5	0.489
**ALP**, *U/L*	87.6 ± 35.7	80.5 ± 24.8	0.127
**Phosphorus**, *mg/dL*	3.69 ± 0.9	3.76 ± 0.64	0.548
**Calcium**, *mg/dL*	10.46 ± 9.62	9.45 ± 0.46	0.986
**Sodium**, *mmol/L*	139.0 ± 2.6	139.2 ± 2.4	0.585
**Potassium**, *mmol/L*	4.04 ± 0.4	4.06 ± 0.32	0.733
**Chloride**, *mmol/L*	105.1 ± 3.1	104.7 ± 2.8	0.376
**Uric acid**, *mg/dL*	5.0 ± 1.8	5.2 ± 1.6	0.375
**γ-GT**, *U/L*	29.2 ± 18.7	39.5 ± 47.6	0.414
**CK**, *U/L*	87.2 ± 61.1	106.2 ± 102.0	0.136
**CRP**, *mg/dL*	0.55 ± 1.58	1.28 ± 10.01	0.535
**hs-CRP**, *mg/dL*	0.89 ± 2.3	0.51 ± 1.17	0.143

Values are expressed as mean ± SD. PSF group: patients with post-stroke fatigue; non-PSF group: patients without post-stroke fatigue. BUN: blood urea nitrogen; AST: aspartate aminotransferase; ALT: alanine transaminase; ALP: alkaline phosphatase; γ-GT: gamma-glutamyl transferase; CK: creatine kinase; CRP: c-reactive protein; hs-CRP: high sensitivity C-reactive protein. ^a^
*t*-test used for total protein, albumin, ALP, phosphorus, sodium, potassium, and uric acid, and Mann–Whitney U test used for other continuous variables.

**Table 5 brainsci-11-01642-t005:** Comparison of blood lipid data and endocrinologic laboratory data between the PSF and non-PSF groups.

	PSF (*n* = 85)	Non-PSF (*n* = 131)	*p*-Value ^a^
**Total Cholesterol**, *mg/dL*	151.4 ± 47.0	150.0 ± 55.8	0.457
**Triglyceride**, *mg/dL*	139.2 ± 107.9	126.6 ± 79.4	0.529
**LDL-Cholesterol**, *mg/dL*	75.8 ± 34.9	74.4 ± 36.7	0.605
**HDL-Cholesterol**, *mg/dL*	55.9 ± 31.8	59.0 ± 28.7	0.063
**Apolipoprotein A1**, *mg/dL*	105.6 ± 16.5	116.2 ± 21.8	<0.001
**Apolipoprotein B**, *mg/dL*	75.6 ± 27.0	73.5 ± 25.4	0.665
**Total lipid**, *mg/dL*	456.9 ± 166.3	440.5 ± 117.5	0.884
**Phospholipid**, *mg/dL*	170.2 ± 36.7	172.3 ± 36.8	0.649
**Homocysteine**, *μmol/L*	12.31 ± 7.49	11.66 ± 4.99	0.884
**HbA1c**, *%*	6.26 ± 1.38	6.48 ± 4.60	0.761
**TSH**, *mIU/L*	2.21 ± 2.05	2.38 ± 1.83	0.314

Values are expressed as mean ± SD. PSF group: patients with post-stroke fatigue; non-PSF group: patients without post-stroke fatigue. LDL: low-density lipoprotein; HDL: high-density lipoprotein; HbA1c: hemoglobin A1c; TSH: thyroid-stimulating hormone. ^a^
*t*-test used for apolipoprotein A1, and Mann–Whitney U test used for other continuous variables.

**Table 6 brainsci-11-01642-t006:** Comparison of clinical features and pattern identification between the PSF and non-PSF groups.

	PSF (*n* = 85)	Non-PSF (*n* = 131)	*p*-Value ^a^
**Insomnia**, *n (%)*	20 (23.5)	32 (24.4)	0.880
**Complexion**, *n (%)*			
Pale	11 (12.9)	21 (16.0)	0.532
Reddened	5 (2.3)	24 (18.3)	0.880
**Faint low voice**, *n (%)*	17 (20.0)	25 (19.1)	0.868
**Thirst**, *n(%)*	10 (11.8)	18 (13.7)	0.673
**Reversal cold of the extremities**, *n(%)*	17 (20.0)	13 (9.9)	0.036
***Pulse***, *n(%)*			
Floating pulse	30 (35.3)	45 (34.4)	0.887
Deep pulse	11 (12.9)	8 (6.1)	0.083
Slow pulse	3 (3.5)	2 (1.2)	0.339
Rapid pulse	2 (2.4)	6 (4.6)	0.397
**Fire-heat pattern**, *%*	9.0 ± 18.7	16.7 ± 26.8	0.014
**Phlegm dampness**, *%*	3.8 ± 11.3	3.4 ± 10.3	0.775
**Qi deficiency**, *%*	27.5 ± 33.5	26.6 ± 35.8	0.856
**Yin deficiency**, *%*	4.9 ± 6.3	5.4 ± 7.7	0.659

Values are expressed as mean ± SD or number (%). PSF group: patients with post-stroke fatigue; non-PSF group: patients without post-stroke fatigue. ^a^ Chi-square test used for clinical features and Mann–Whitney U test used for pattern identification.

**Table 7 brainsci-11-01642-t007:** Logistic regression results for predicting post-stroke fatigue.

Patient Characteristic	β Coefficient	95% CI		*p*-Value
**Fire-heat pattern**	−0.003	−0.006	0.000	0.028
**Apolipoprotein A1**	−0.005	−0.009	−0.002	0.003
**aPTT**	0.015	−0.001	0.030	0.066
**Age**	0.005	−0.001	0.010	0.088

CI: confidence interval; R^2^ = 0.122. aPTT: activated partial thromboplastin time.

## Data Availability

The data presented in this study are available on request from the corresponding author. Because the data is based on medical records, the data are not publicly available.

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
