# Peer review of "An Analysis of Characteristics of Post-Stroke Fatigue in Patients without Depression: A Retrospective Chart Review"

_brainsci, 2021, doi:10.3390/brainsci11121642_

Round 1

Reviewer 1 Report

The authors aim to evaluate post-stroke fatigue (PSF) which is an important contributor to delayed/poor post-stroke recovery. This is a retrospective study based on medical records of stroke patients without depression. Biochemical profile was assessed in controls and study group in addition to PSF-related observations. Higher occurrence of cold extremities and less fire-heat pattern was observed in PSF group and lower levels of Apolipoprotein A1 were found in the PSF group. Overall, the study is relevant to the current challenges of post-stroke recovery.

Comments:

Introduction – Please add further description about PSF e.g. symptomatology, major complaints, diagnosis etc. Also, please provide more details on how PSF possibly interferes with recovery and rehabilitation?

Methods - The control subjects are stroke patients without fatigue and not stroke patients with depression. Please also mention this in the abstract.

In study subjects, non-depressed patients were identified based on assessment done at the time of admission to the hospital. Can authors explain further what are they trying to imply when they say patients without depression – pre-existing depression, depression just before admission but after stroke or depression observed after stroke during recovery?

Please briefly include what parameters/features does the FAS and FSS scale measure?

What is the significance of religion as an assessment marker in demographics? Are there prior studies that identify incidence of PSF in certain religious populations?

Results - ApoA1 levels were measured as a routine analyte in the overall lipid profile. previously, ApoA1 levels have been identified as a prognostic biomarker for acute ischemic stroke. Did the authors observe any stroke type-specific differences in the ApoA1 levels? Please include this in the discussion as well.

Despite statistically significant, the mean + SD of ApoA1 levels is within close range of each other. Authors should present this data (and wherever possible) as box plots for better data visualization. Also, if data permits, a ROC curve should be made to assess AUC for use of ApoA1 as a biomarker of PSF.

Reviewer 2 Report

The authors have conducted a retrospective study aimed at assessing the implication of some factors in post-stroke fatigue without depression. The authors found that ApoA1 levels were significantly lower in PSF than in the control group. There is also a signnificant increase of reversal cold of the extremities in  PSF.

Broad comment:

The manuscript is well written. The topic of this work is of relevance and interest. The hypothesis of work is reasonable and justified. Results are appropriately described.

Minor points:

It would be relevant the authors address the anti-inflammatory role of ApoA1 in their discussion.  In fact, the authors have recently published a paper (https://doi.org/10.3390/healthcare9111586) addressing the role of some inflammatory markers. It would be worthy they discuss the present results taking into account the ones of the previoulsy published paper.
